# *Lactobacillus plantarum* J26 Alleviating Alcohol-Induced Liver Inflammation by Maintaining the Intestinal Barrier and Regulating MAPK Signaling Pathways

**DOI:** 10.3390/nu15010190

**Published:** 2022-12-30

**Authors:** Hongxuan Li, Shasha Cheng, Jiacheng Huo, Kai Dong, Yixin Ding, Chaoxin Man, Yu Zhang, Yujun Jiang

**Affiliations:** Key Lab of Dairy Science, Ministry of Education, College of Food Science, Northeast Agricultural University, Harbin 150030, China

**Keywords:** alcoholic liver disease, probiotics, MAPK, gut microbiota, intestinal barrier

## Abstract

Alcoholic liver disease (ALD), as a global health problem, is mainly caused by liver inflammation. Meanwhile, probiotics have been considered as a potential and promising strategy to prevent and alleviate ALD. This study aimed to investigate the ameliorative effect of pre-intaking with *Lactobacillus plantarum* J26 (*L. plantarum* J26) on alcohol-induced liver inflammation, with emphasis on the underlying mechanism for alleviating ALD. The results indicated that *L. plantarum* J26 could reduce the abundance of Gram-negative pathogenic bacteria by regulating the gut microbiota in mice with alcoholic liver injury, thereby reducing the lipopolysaccharide (LPS) content in the intestine. In addition, *L. plantarum* J26 could also maintain the intestinal barrier, prevent LPS from crossing the intestinal barrier to correct disorders of the gut–liver axis and then inhibit the activation of Toll-like receptor 4 (TLR4)-mediated MAPK signaling pathway, reducing liver inflammation and restoring liver functions. In conclusion, pre-intake of *L. plantarum* J26 could alleviate alcohol-induced liver inflammation, which may be closely related to the role of intestinal microbiota in regulating and maintaining the intestinal barrier and then regulating the MAPK signaling pathway.

## 1. Introduction

Alcohol is widely consumed worldwide, and people often get sick from drinking alcohol to excess. According to statistics, more than 2 billion people worldwide regularly drink alcohol, and approximately 3 million people die from excessive alcohol consumption each year, accounting for 6% of all deaths worldwide [1]. Alcoholic liver disease (ALD) is the chronic disease with the highest morbidity and mortality in the world, and alcohol exposure is strongly associated with ALD [2]. ALD includes alcoholic fatty liver, alcoholic steatohepatitis, alcoholic hepatitis and alcoholic fibrosis, which may lead to hepatocellular carcinoma (HCC) [3]. These diseases may also coexist in the same individual and occur sequentially; women are more likely to suffer from them than men, causing great damage to the human body [4].

Recent studies have found that the occurrence of ALD is closely related to the gut and its microbiota [5]. Ingestion of alcohol as it is digested in the gut can affect the gut microbiota, thereby increasing the abundance of Gram-negative bacteria and producing more lipopolysaccharide (LPS). Excessive alcohol consumption also increases intestinal permeability, causing that more LPS cross the intestinal barrier and even reach the liver through the intestinal–liver axis, where LPS can reach the liver and activate TLR4, which activate Kupffer cells through signal transduction. When the above pathways are activated, Kupffer cells begin to release reactive oxygen species (ROS), adhesion molecules, chemokines, and proinflammatory cytokines, which eventually induce liver inflammation and severely impair liver health [6,7,8].

In recent years, more and more studies have confirmed the role of probiotics in regulating the gut microbiota and maintaining the intestinal barrier [9,10,11]. At the same time, probiotics regulation of gut microbiota to alleviate alcohol-induced liver inflammation is a promising potential therapeutic approach [12,13]. *Lactobacillus plantarum* J26 (*L. plantarum* J26) is a probiotic strain isolated from Chinese traditional fermented dairy products with good probiotic efficacy. Previous studies have shown that *L. plantarum* J26 alleviates LPS-induced liver injury and has immunomodulatory effects. In addition, in vitro studies have shown that fermented blueberry juice of *L. plantarum* J26 alleviates oxidative stress [14,15,16], but the potential of *L. plantarum* J26 to alleviate alcohol-induced liver inflammation still remains to be explored.

This study was designed to investigate the effects of *L. plantarum* J26 on liver inflammation, gut microbiota, intestinal barrier and MAPK signaling pathway-related protein expression in mice with acute-on-chronic alcoholic liver injury. To provide evidence for laboratory prevention of liver inflammation induced by alcohol.

## 2. Materials and Methods

### 2.1. Materials, Strain, and Media

*L. plantarum* J26 (also known as *L. plantarum* NDC75017 [14], *L. plantarum* TD109 [17], *L. plantarum* LJ26) was isolated from traditional Chinese fermented dairy products. The strain was used after incubation in Man Rogosa Sharp broth (MRS, Hope Bio-Technology Co., Ltd., Qingdao, China) at 37 °C for 18 h.

### 2.2. Animal Experiment and Its Parameter Analysis

#### 2.2.1. Animal and Experimental Design

The male C57BL/6J mice (*n* = 50, 6 weeks) were from Vital River Laboratory Animal Technology Co., Ltd., (Beijing, China). The animal experiments in this study were allowed by the Laboratory Animal Welfare and Ethics Committee of Northeast Agricultural University (#NEAU-2021-06-0286-32). A mice acute-on-chronic alcoholic liver injury model was established with simple modifications based on Gao-Bing’s method [18]. Briefly, the first week of adaptive feeding alcohol concentration was gradually increased from 0% to 5% (*v*/*v*) and this concentration was maintained for feeding (chronic phase), then high concentration (31.5% (*v*/*v*)) alcohol feeding was given (acute phase) the day before execution. A total of 50 male C57BL/6J mice (6 weeks old) with a body weight of 20–25 g were selected and kept in an animal room with a temperature control of 22 ± 2 °C. The mice were adaptively fed with standard feed for 1 week. They were randomly divided into five groups (normal control (NC), Alcohol, low-dose *L. plantarum* J26 (LLP), medium-dose *L. plantarum* J26 (MLP), and high-dose *L. plantarum* J26 (HLP); 10 mice per group) and were given Lieber–DeCarli alcohol liquid diet [19] (Teluofei Feed Technology Co., Ltd., Nantong, China) transition adaptive feeding for 1 week (alcohol concentration from 0 to 5% (*v*/*v*)), and the NC group was given an isocaloric control liquid diet feed.

After the transition period, the Alcohol group, LLP group, MLP group and HLP group continued to be fed with alcohol liquid feed for 8 weeks (alcohol concentration 5% (*w*/*v*)) of which the mice of the Alcohol group were given 0.2 mL sterile PBS every morning; LLP 0.2 mL low-dose *L. plantarum* J26 (1 × 10^7^ CFU/mL), medium-dose *L. plantarum* J26 (1 × 10^8^ CFU/mL) and high-dose *L. plantarum* J26 (1 × 10^9^ CFU/mL) were administered orally every morning in LLP group, MLP group and HLP group, respectively. High-concentration alcohol (31.5% (*v*/*v*)) intragastric gavage was given at approximately 7:00 the next morning on the 8th weekend; the dose was 5 g/kg, and the mice were killed after fasting for 9 h. NC group was fed with special control maltodextrin liquid diet and other calories for 8 weeks (the daily feeding amount was the average dietary amount of the other 4 groups on the previous day), and 0.2 mL sterile PBS was gavage every morning. Maltodextrin equivalent to a high concentration of alcohol (31.5%, *v*/*v*) calories was given by gavage at 7 a.m. the following day at the end of week 8 at a dose of 5 g/kg. The scheme of animal experiment is shown in Figure 1.

#### 2.2.2. Body Weight and Liver Index

The clinical manifestations and feeding situation of the mice in each group were observed and recorded each day, and each mice were weighed each week. During the necropsy, the conditions of the liver and intestine were observed, and the weight of the liver was recorded. The ratio of the weight of the liver to the body weight was the liver index.

#### 2.2.3. Histomorphological Observation

##### Anatomical Observation

Liver tissue was obtained at low temperature and placed on ice for anatomical observation of it. Liver size, color and margins were observed and recorded under bright light conditions.

##### Histopathological Examination

Fresh liver and colon tissues were immediately fixed in 10% formaldehyde solution for 48 h, followed by routine paraffin embedding and sectioning. At room temperature, the sections were immersed in the following reagents for deparaffinization and hydration: xylene I for 10 min, xylene II for 10 min, absolute ethanol for 5 min, 95% ethanol for 5 min, 80% ethanol for 5 min, and 70% ethanol for 5 min, double distilled water for 5 min. After completion, the instructions for hematoxylin-eosin staining (HE staining) were followed. Histological images were taken by the microscopy imaging system (Leica DM1000, Nussloch, Germany).

#### 2.2.4. Biochemical Assays

The activities of alanine aminotransferase (ALT), aspartate aminotransferase (AST), and alkaline phosphatase (ALP) in serum were determined by an automatic biochemical analyzer (Shandong Broke Biological Industry Co., Ltd., Shandong, China).

The levels of triglyceride (TG), total cholesterol (TC) in the serum of the mice were measured using commercial assay kits (Nanjing Jiancheng Bioengineering Institute, Nanjing, China) according to the instructions of the manufacturer.

The levels of tumor necrosis factor-α (TNF-α), interleukin-1β (IL-1β), interleukin-6 (IL-6) and interleukin-10 (IL-10) in the mouse livers, and the concentration of LPS in the mice serum were measured with enzyme-linked immunosorbent assay (ELISA) kits (Quanzhou Kenuodi Biotechnology Co., Ltd., Quanzhou, China), following the instructions of the manufacturer.

#### 2.2.5. Immunohistochemical

The liver tissue fixed in 10% methanol solution was embedded in paraffin and sectioned. After the sections were dewaxed and hydrated, myeloperoxidase (MPO) was determined as follows.

(1)The antigen was repaired with citrate antigen repair solution by microwave method and then cooled to room temperature;(2)A total of 3% H_2_O_2_ was added and incubated for 15 min at room temperature and then washed three times;(3)A total of 3% goat serum was added and blocked for 1 h at room temperature;(4)Diluted MPO antibody was added and incubated at 4 °C overnight;(5)Washed with PBS three times, 5 min/time. The biotinylated secondary antibody was added and incubated for 1 h at room temperature and then washed with PBS again three times;(6)After adding peroxidase-labeled streptavidin, all sections were incubated at 37 °C for 30 min and then washed with PBS three times;(7)3,3′-diaminobenzidine tetrahydrochloride (DAB) chromogenic solution was added and incubated for 1–5 min for hematoxylin counterstaining;(8)After dehydration with different concentrations of ethanol solution and diaphanization with xylene, the sections were sealed with neutral gum, and air-dried;(9)Placed under a microscope, observed, and recorded.

#### 2.2.6. RNA Extraction and RT-Quantitative PCR (RT-qPCR)

RNA from the liver was extracted with a Simple P Total RNA Extraction Kit (Bioer Technology Co., Ltd., Hangzhou, China). NanoDrop spectrophotometer (Thermo, Wilmington, DE, USA) was used to measure the purity and integrity of RNA samples. A reverse transcription kit (Takara Bio, Shiga, Japan) was used for reverse transcription. Additionally, RT-qPCR was performed by SYBR Premix Ex Taq (Takara Bio, Shiga, Japan) according to the manufacturer’s instructions, the reaction was conducted with a QuantStudio™ 3 system. Two sets of specific primers were designed by Primer 5.0 software for the experiment (Table 1). The relative gene expression was analyzed using the 2^−∆∆Ct^ method, where Ct is the threshold cycle. The β-actin encoding gene was used as the reference gene in mice.

#### 2.2.7. Western Blotting Analysis

Liver tissues were homogenized on ice in ice-cold radioimmunoprecipitation assay (RIPA) lysis buffer (KeyGEN BioTECH, Nanjing, China) supplemented with 0.1% protease inhibitor, 1% phosphatase inhibitor, and 0.5% phenylmethanesulfonyl fluoride. The liver homogenate was then centrifuged to obtain the total protein. The nuclear protein and cytoplasmic protein were harvested with commercial kits (Invent Biotechnologies, Inc., Beijing, China).

The related protein expression levels were measured by Western blotting. The antibodies, TLR4 (A5258), p38 (A4771), *p*-p38 (AP0526), JNK (A5051), *p*-JNK (AP0275), ERK (A4782) and *p*-ERK (AP0485), used in the experiments were purchased from Wuhan Aibotec Biotechnology Co., Ltd. Whole proteins were extracted from liver tissue according to the operation of the Invent Kit and tissue protein concentrations were measured using the bicinchoninic acid (BCA) protein concentration Assay Kit (KeyGEN Biotech, Nanjing, China). After sodium dodecyl sulfate-polyacrylamide gel electrophoresis (SDS-PAGE) electrophoresis, the membrane was transferred at 100 V constant pressure for 1–2 h and then blocked at room temperature for 1 h. After removing the blocking solution, the diluted primary antibody was added at 4 °C overnight. The next day, the secondary antibody was washed with tris buffered saline with Tween-20 (TBST) 3 times, 5 min/time. The secondary antibody was continued to be added and incubated at room temperature for 30 min and then washed with TBST 4 times. Image J software processing system was used to analyze the optical density value of the target band.

#### 2.2.8. Metagenomic Analysis of Gut Microbiome

Total DNA was extracted according to the instructions of the fecal DNA extraction kit. The concentration and purity of DNA were detected by Quick Drop. V3 and V4 regions of 16S rDNA were used as target sequences for amplification. The PCR products were detected by agarose gel electrophoresis. Illumina MiSeq sequencer was used for on-machine sequencing. Greengenes database was used for comparison.

### 2.3. Statistical Analysis

SPSS software was used for one-way ANOVA analysis. Excel software and GraphPad Prism 8.02 (GraphPad Software, San Diego, CA, USA) were used for plotting. All tests were repeated at least three times, and results were presented as mean ± standard deviation (SD).

## 3. Results

### 3.1. Effects of L. plantarum J26 on Body Weight and Liver Index in the Mice

Figure 2 shows the body weight and liver index of the mice in each group during the study. There was no significant difference in the body weight of the mice at the beginning of the test (*p* > 0.05). The body weight of the mice in each group gradually increased from 1 to 4 weeks, and then stabilized after the 8th week. The HLP group was the closest to the NC group.

The liver index can directly reflect liver injury. In terms of liver index, the LLP and MLP groups were significantly smaller than the alcohol group (*p* < 0.05), and the NC and HLP groups were significantly smaller than those in the Alcohol group (*p* < 0.01), indicating that *L. plantarum* J26 can relieve alcohol-induced liver congestion and edema. These results indicated that *L. plantarum* J26 could alleviate liver congestion and edema caused by alcohol.

### 3.2. Effects of L. plantarum J26 on Liver Histopathology in Liver of Mice

The liver of mice was observed anatomically, and the results are shown in Figure 3a. The liver in the NC group was reddish-brown, with a soft texture, and the borders of each lobule were demarcated. The livers of mice were swollen and light in color after alcohol induces. After *L. plantarum* J26 gavage, the shape, size and color of the liver were restored to varying degrees.

Histopathological observation of mice liver tissue slices was obtained by H&E staining. As shown in Figure 3b, the liver tissue structure of the mice in the NC group was clear, the hepatic lobules were intact, the hepatocytes were arranged regularly, and there was no swelling and deformation. Compared with the NC group, the liver tissue structure of the mice in the Alcohol group was severely abnormal, most of the hepatocytes were fatty and edema, the cells were swollen, the cytoplasm was lightly stained, and some inflammatory cells were infiltrated. The liver tissue structure of mice in the LLP and MLP groups existed abnormal, some hepatocytes were slightly edematous, cytoplasmic vacuolation, and inflammatory cell infiltration still existed, but the degree of liver injury was significantly improved compared with the Alcohol group. The HLP group had very few lipid droplets in the liver tissue, and no obvious inflammatory cell infiltration was found. This indicated that a high dose of *L. plantarum* J26 could more effectively reduce the pathological damage of liver tissue.

### 3.3. Effects of L. plantarum J26 on the Levels of Liver Function Enzymes, LPS and Lipid Content in the Serum of Mice

The serum levels of liver function enzymes activity and LPS content are shown in Figure 4. Compared with the NC group, alcohol induction significantly increased the activities of liver function enzymes (*p* < 0.001), and the ratio of AST to ALT in the Alcohol group was greater than 2, suggesting severely impaired liver function. After pre-intake of *L. plantarum* J26, liver function enzymes activities showed a decreasing trend in all groups of mice. Compared with the Alcohol group, the activities of ALT, AST, and ALP in the LLP group were decreased by 10.4% (*p* < 0.05), 13.1% (*p* < 0.01), and 3.7%, respectively; the activities of ALT, AST, and ALP in the MLP group decreased by 16.7% (*p* < 0.01), 27.6% (*p* < 0.001), and 8.31% (*p* < 0.05), respectively; the activities of ALT, AST, and ALP in the HLP group decreased by 44.61%, 58.8%, and 16.8%, respectively, all showing significant differences (*p* < 0.001). It indicates that *L. plantarum* J26 could effective in improving impaired liver function in a dose-dependent manner.

As shown in Figure 4d, the serum LPS content of mice in the Alcohol group was significantly higher than that in the NC group (*p* < 0.01). Compared with the Alcohol group, the serum LPS content in the MLP group was significantly reduced by 51.4% (*p* < 0.05); the serum LPS content in the HLP group was significantly reduced by 60.1% (*p* < 0.01). There was no significant reduction in the content of serum LPS in the LLP group may be due to the lower dose of 1 × 10^7^ CFU/mL *L. plantarum* J26. The results showed that high doses of *L. plantarum* J26 could reduce the LPS content in mouse serum more effectively.

TG and TC are the main forms of lipid accumulation. The TG and TC concentration in the liver of all groups of mice is represented in Table 2. Compared with the NC group, the content of TG and TC in the liver tissue of the mice in the Alcohol group was significantly increased (*p* < 0.05 and *p* < 0.01). Different doses of *L. plantarum* J26 could decrease the serum lipid content to different degrees compared with the Alcohol group. These results indicated that *L. plantarum* J26 could reduce alcohol-induced lipid accumulation.

### 3.4. Effect of L. plantarum J26 on the Levels of MPO and Inflammatory Factors in the Liver of Mice

The degree of hepatic neutrophil infiltration in the liver is generally positively correlated with the severity of liver inflammation. MPO is a functional marker and activation marker of neutrophils, and inflammatory factor levels may also reflect levels of liver inflammation. In this study, immunohistochemistry was used to analyze hepatic neutrophil infiltration, and ELISA was used to detect hepatic levels of pro-inflammatory factors TNF-α, IL-1β, IL-6 and anti-inflammatory factor IL-10.

As shown in Figure 5, the number of MPO-positive cells was significantly increased in the liver of mice in the Alcohol group compared with the NC group (*p* < 0.001). Compared with the Alcohol group, the number of MPO-positive cells in the liver of the LLP, MLP, and HLP groups was decreased by 19.2% (*p* < 0.05), 30.1% (*p* < 0.01), and 36.2% (*p* < 0.001), showing a certain dose-dependent effect.

As shown in Figure 6, the levels of pro-inflammatory factors TNF-α, IL-1β, and IL-6 were significantly increased in the Alcohol group compared with the NC group (*p* < 0.01, *p* < 0.001, *p* < 0.001). Compared with the Alcohol group, pro-inflammatory factors were mostly significantly regulated by HLP (*p* < 0.01), with moderately effected by the MLP and LLP groups especially in IL-6 and IL-10 (*p* < 0.05). Levels of three pro-inflammatory factors in the HLP groups was decreased by 35.4% (*p* < 0.01), 34.3% (*p* < 0.001) and 31.14% (*p* < 0.001) compared with the Alcohol group, respectively. The levels of the anti-inflammatory factor IL-10 was reduced to 60.7 ± 8.3 ng/L (*p* < 0.001) in the Alcohol group compared with the NC group. Compared to the Alcohol group, pre-intake of *L. plantarum* J26 significantly increased IL-10 levels to 94.5 ± 7.9 ng/L. (*p* < 0.01). All the above results indicated that different doses of *L. plantarum* J26 could reduce the level of liver inflammation in mice to different degrees in a dose-dependent manner.

### 3.5. Effects of L. plantarum J26 on Protein Expression Related to MAPK Signaling Pathway in Mice Liver

The expression of TLR4 protein and related proteins (p38, JNK, ERK) in MAPK signaling pathway and their phosphorylation levels were detected to explore the mechanism by which *L. plantarum* J26 inhibits inflammatory response. The results were shown in Figure 7, compared with the NC group, TLR4 protein expression and phosphorylation levels of p38, JNK and ERK proteins were significantly up-regulated in the Alcohol group (*p* < 0.001), indicating that the MAPK signaling pathway was activated. However, pre-intake of *L. plantarum* J26 at all doses could significantly reduce the expression of TLR4 protein in a dose-dependent manner. Compared with the Alcohol group, pre-intake of *L. plantarum* J26 at all doses significantly down-regulated the phosphorylation level of p38 protein (*p* < 0.01, *p* < 0.001 and *p* < 0.001); the phosphorylation level of JNK protein was significantly down-regulated in the MLP and HLP groups (*p* < 0.05); the phosphorylation level of ERK protein was significantly down-regulated in the MLP and HLP groups (*p* < 0.05 and *p* < 0.01); the phosphorylation levels of three proteins were decreased in the LLP group, but the phosphorylation levels of JNK and ERK proteins were not significantly different. In conclusion, *L. plantarum* J26 could alleviate liver inflammation by inhibiting the TLR4-mediated MAPK signaling pathway.

### 3.6. Protective Effects of L. plantarum J26 on Alcohol-Induced Intestinal Injury in Mice

H&E staining was used to observe the histopathological changes of colon tissue in mice, and the results were shown in Figure 8a. The epithelial cells in the mucosal layer of the NC group of mice were neatly arranged, with dense tubular glands in the lamina propria, with intact structure and undiminished number, and no inflammatory infiltration was seen. In the Alcohol group, the local myofibrillar gap was enlarged, the muscle layer was loosely arranged, and the submucosa was moderately edematous.

When alcohol enters the mice, it is absorbed into the intestine, greatly increasing the likelihood of intestinal barrier disruption in mice. Therefore, the gene expression levels of tight junction-related proteins Claudin-1, Occludin, and ZO-1 were measured. The results were shown in Figure 8b–d, compared with the NC group, the gene expression levels of the three related tight junction proteins in the intestine of the Alcohol group were significantly downregulated, including Claudin-1 downregulated by 17.5% (*p* < 0.05), Occludin downregulated by 55.2% (*p* < 0.001) and ZO-1 downregulated by 14.0% (*p* < 0.01). Compared with the Alcohol group, the expression of three related tight junction proteins was significantly upregulated in the HLP group, including Claudin-1 expression upregulated by 17.6% (*p* < 0.05), Occludin expression upregulated by 28.8% (*p* < 0.05), and ZO-1 expression upregulated by 9.3% (*p* < 0.05). These results suggested that high-dose *L. plantarum* J26 could more effectively regulate the transcription level of tight junction-related protein genes in the colon, improve the tight junction structure of the intestinal barrier, and alleviate intestinal injury in mice.

### 3.7. Effects of L. plantarum J26 on the Gut Microbiota in Mice

In recent years, more and more studies have found that changes in gut microbiota are strongly related to the development of alcoholic liver injury [11]. Therefore, this study detected and analyzed the effects of different doses of *L. plantarum* J26 on the gut microbiota of mice with alcoholic liver injury. A total of 2,103,159 original sequences were obtained from 20 samples by 16SrDNA high-throughput sequencing, 120,504 low-quality sequences were removed, 1,926,321 sequences were obtained after denoising, and the final 1,505,272 high-quality sequence were obtained after splicing and chimera removal, which could satisfy the requirements of the test samples.

#### 3.7.1. Gut Microbiota Diversity in Mice

The effects of alcohol and *L. plantarum* J26 on gut microbiota was evaluated by gut microbiota Alpha diversity analysis. Chao1 was usually used to estimate the total number of species, Shannon and Simpson’s index represents the diversity of the community, with Shannon being more sensitive to species richness and Simpson more sensitive to species evenness. The results are shown in Figure 9a, compared with the NC group, the mice in the Alcohol group showed a decrease in all three indexes, but there was no difference between the groups. Pre-intake different doses of *L. plantarum* J26 could improve Chao1, Shannon, and Simpson indexes in alcohol-induced mice. The above results showed that *L. plantarum* J26 could restore the richness and diversity of alcohol-induced gut microbiota in mice to a certain extent.

The Beta diversity of microbiota indicates the differences in microbiota composition between samples. In this study, the principal coordination analysis (PCoA) and non-metric multidimensional scaling (NMDS) analysis based on the Bray-Curtis distance matrix were used to calculate the Beta diversity of the sample microbiota. The results of PCoA analysis were shown in Figure 9b, the distance between the Alcohol group and other groups was far and distinguished. The distances within and between groups of different doses of *L. plantarum* J26 decreased distances were similar to those of the NC group, indicating that pre-intake of *L. plantarum* J26 could improve the disorder of gut microbiota and the stability and similarity of gut microbiota structure in mice after alcohol induction. When NMDS analysis was performed, the stress values of the results were less than 0.2, indicating reliable results. As shown in Figure 9c, the stress value of this test was 0.117, which could accurately reflect the difference in microbiota composition between samples. It can also be seen that the Alcohol group was significantly separated from the other groups, indicating that the induction of alcohol seriously altered the structure of gut microbiota. However, the similar distances between the groups of pre-intake *L. plantarum* J26, especially in the HLP group and NC group, suggested that pre-intake of *L. plantarum* J26 stabilizes the gut microbiota structure.

#### 3.7.2. Analysis of Microbial Colony Composition at the Phylum Level

In order to further investigate the differences in the effects of *L. plantarum* J26 and alcohol on the structure of gut microbiota of mice, the gut microbiota was classified at different levels and then represented by graphs. The species composition and abundance of gut microbiota of different groups of mice were first analyzed at the phylum level. As shown in Figure 9a, there were significant differences in the proportion and abundance of gut microbiota in different groups of mice. Overall, *Firmicutes*, *Actinobacteria* and *Proteobacteria* accounted for approximately 80% of the dominant bacteria in the gut at the phylum level. In addition, *Bacteroidetes* and *Verrucomicrobia* ranked in the top five in terms of microbiota abundance.

Compared with the NC group, the abundance of Firmicutes and *Verrucomicrobia* in the gut of mice in the Alcohol group was significantly decreased by 24.3% and 1.1%, respectively, while the abundance of *Actinobacteria* and *Proteobacteria* increased significantly by 22.7% and 0.5%, respectively. The abundance of *Bacteroidetes* was also increased, indicating that alcohol induced an imbalance in the gut microbiota of mice. Compared to the Alcohol group, the abundance of *Firmicutes* in the intestinal tract of *L. plantarum* J26 group was increased at all doses, and the high dose of pre-intake protected *Firmicutes* abundance to 62.0%. At the same time, it could significantly reduce the abundance of *Proteobacteria* and maintain it at 2.2%, thus reducing the growth of harmful intestinal bacteria. However, due to the complex effect of Lactobacillus on the gut microbiota, further genus-specific dependence level analysis is required.

#### 3.7.3. Analysis of Microbial Colony Composition at the Genus Level

As shown in Figure 10b, at the genus level, compared to the NC group, the abundances of *Allobaculum*, *Lactobacillus*, *Corynebacterium*, *Oscillospira*, *Akkermansia*, *Parabacteroides*, *Streptococcus* and *Bifidobacterium* in the intestinal tract of mice in the Alcohol group were decreased, while the abundances of these bacteria were increased at all doses in the *L. plantarum* J26 group. In addition, the abundance of *Lactobacillus* in the Alcohol group increased from 10.9% to 13.9% in the HLP group, which was close to that in the NC group, indicating that *L. plantarum* J26 could well supplement or increase the abundance of *Lactobacillus* in mice. In addition, compared with the NC group, the abundance of *Sutterella* and *Aerococcus* was increased in the Alcohol group, while it was decreased in all dose groups. Compared to the NC group, *Clostridium* abundance was reduced in the Alcohol group and *L. plantarum* J26 groups. The abundance of the remaining genera did not vary significantly among the groups.

## 4. Discussion

There is no doubt that alcoholic liver injury is closely related to the occurrence and development of liver inflammation [8]. Modulation of the gut microbiota through ingestion of probiotics has attracted increasing attention as a potentially promising therapeutic approach to alleviate ALD [5]. In this study, pre-intake of *L. plantarum* J26 could reduce the number of Gram-negative pathogenic bacteria, such as *Proteobacteria* and *Bacteroides*, in the gut microbiota by restoring alcohol-induced gut microbiota disturbed and improved the intestinal barrier to prevent LPS from reaching the liver through the gut-liver axis. In addition, while decreasing the level of LPS in the liver, it would also inhibit the activation of TLR4-mediated MAPK signaling pathway, reducing the inflammatory response and achieving hepatoprotection. These results suggest that *L. plantarum* J26 had a good effect on alcohol-induced liver inflammation, and its related mechanism is consistent with the results of some previous studies.

The activities of ALT, AST, and ALP are often used clinically as indicators to evaluate liver function, and their levels can reflect the degree of liver injury within a certain range [20]. In addition, anatomical observation and H&E staining can more directly reflect the development of liver tissue inflammation on mouse. The results showed that alcohol-induced increased serum ALT, AST, and ALP activities, abnormal liver structure, hepatocyte steatosis, edema, and inflammatory cell infiltration. Pre-intake of *L. plantarum* J26 significantly inhibited alcohol-induced increase in ALT, AST, and ALP activities in a dose-dependent manner. At the same time, pre-intake of *L. plantarum* J26 also improved alcohol-induced liver pathological changes, and the morphological structure of liver tissue in the HLP group was the most similar to that in the NC group. Kirpich et al. [21] reduced ALT and AST activity in serum of patients with ALD by administering *Bifidobacterium bifidum* and *Lactobacillus plantarum* 8PA3 Da et al. [22] used lactic acid bacteria to ferment cress, and the products obtained could also inhibit the increase of ALT, AST, and ALP activities in the serum of alcohol-exposed mice, and ameliorated alcohol-induced histopathological alterations in the liver. Based on the enzymatic evaluation of liver function, anatomical and pathological observations, it was tentatively concluded that *L. plantarum* J26 had a significant improvement effect on alcoholic liver injury.

Activation of Kupffer cells is one of the key cellular events in the pathogenesis of ALD. After alcohol intake, increased intestinal permeability and hepatocellular damage lead to the release of pathogen- or damage-associated molecular patterns (PAMPs/DAMPS), which in turn activate Kupffer cells via the TLRs pathway. Activated Kupffer cells will secrete a large number of cytokines making hepatic inflammation and ALD worse [23,24,25]. The development of ALD is not only related to proinflammatory factors, such as TNF-α, IL-1β, and IL-6 but also to the level of anti-inflammatory factors, such as IL-10. Studies have shown that IL-10 played an anti-fibrosis role in the Progression of liver inflammation and also played an important regulatory role on TNF-α [26,27]. These inflammatory mediators not only trigger local inflammation in liver tissue, but also activate neutrophils to further aggravate the degree of liver injury, and MPO can be used as a marker of neutrophil activation. The results of this study showed that alcohol intake significantly increased the levels of MPO, TNF-α, IL-1β, and IL-6 and decreased the level of anti-inflammatory factor IL-10 in the liver and that pre-intake of *L. plantarum* J26 significantly improve these conditions. Sangineto et al. [28] used Dimethyl-fumarate to reduce the expression of TNF-α and IL-1β in alcoholic liver injury. Yang et al. [29] alleviated the abnormal levels of TNF-α, IL-6, and IL-10 in mice with alcoholic liver injury by feeding inulin. Dhiman et al. [30] normalized plasma TNF-α, IL-6, and IL-10 levels in patients with alcoholic cirrhosis by feeding probiotic VSL#3. These findings are consistent with the results of the present study, which suggested that pre-intake of *L. plantarum* J26 could alleviate alcohol-induced liver inflammation.

Alcohol intake not only increases intestinal permeability but also increases the production of more LPS by Gram-negative bacteria in the gut microbiota, which crosses the intestinal barrier into the blood and reaches the liver through the portal vein, causing secondary damage to the liver. TLR4 is the specific receptor for LPS, and TLR4 activates the expression of MAPK and other signaling pathways after receiving LPS stimulation. The MAPK pathway has a three-stage signaling process, namely the three-stage kinase pattern: MAPK, MAPK kinase, MAPK kinase, and the three kinases are activated sequentially [31]. P38, JNK, and ERK are three branches of MAPKs that are essential for the expression of pro-inflammatory factors [32]. Ma et al. [33] found that knockdown or inhibition of the p38 gene attenuated the aggravation of neutrophil infiltration and alleviated liver injury and inflammation in an acute-on-slow alcohol-fed mice model, indicating that p38 plays a key role in slow-on-acute alcohol-fed liver injury. The study results of Jiang et al. [34] showed that Poria cocos polysaccharide (PCP-1C) could reduce liver inflammation by inhibiting MAPK signaling pathway. This suggests that the downregulation of this signal may be a way to prevent inflammation in the liver. Therefore, this study examined serum LPS levels, TLR4 protein expression, and the expression of three important kinases in the MAPK signaling pathway. The results showed that LPS content in serum decreased and the TLR4 protein expression and p38, JNK, and ERK protein phosphorylation levels in liver were significantly down-regulated after pre-intake of *L. plantarum* J26, which was consistent with the study of Wang et al. [35]. The results indicated that *L. plantarum* J26 could protect the liver by inhibiting the TLR4-mediated MAPK pathway, reducing the secretion of inflammatory factors and neutrophil infiltration, and alleviating alcohol-induced liver inflammation in mice.

The gut is the body’s largest and most important internal barrier, which protects the host from harmful substances. The intestinal barrier is composed of the mucus layer in the lamina propria, epithelial cells and immune cells, and alcohol is one of the most common toxins that disrupt the integrity of the intestinal barrier and the gut microbiota [36]. Tight junctions are highly diverse structures composed of tight junction proteins that are located at the apex of intestinal epithelial cells and are the primary means of connecting intestinal epithelial cells. Alcohol induction will first reduce the expression of tight junction proteins, destroy the tight junction structure, leads to increased intestinal barrier permeability, and allows LPS to leak in the intestinal lumen and reach the liver through the gut-liver axis, stimulating Kupffer cells and eventually causing liver inflammation and injury [37]. Therefore, in this study, the histopathological observation of colon tissue was first performed, and then the gene expression of tight junction proteins Claudin-1, Occludin, and ZO-1 were detected. The results showed that alcohol-induced colonic tissue myofiber space enlargement, muscle layer arrangement relaxation, submucosal edema, and significantly decreased gene expression of the tight junction proteins Claudin-1, Occludin, and ZO-1. However, pre-intake of *L. plantarum* J26 maintained normal mucosal structure and restored the gene expression of Claudin-1, Occludin, and ZO-1 to some extent. Das et al. [10] found that fed mice osteopontin increased gene expression of intestinal tight junction proteins and maintained intestinal barrier function, thereby alleviating ALD. Wang et al. [38] used chitosan to promote the expression of tight junction proteins Claudin-1, Occludin, and ZO-1 to maintain intestinal barrier function, which is consistent with the results of the present study.

In recent years, there has been an increasing number of studies on the correlation between the gut microbiome and human diseases, and there is evidence that the gut microbiome is strongly associated with alcohol-induced related diseases [39]. Colonization of the intestinal tract by lactic acid bacteria can affect liver tissue diseases. Huang et al. [40] showed that probiotics *Bifidobacterium* (including active *Lactobacillus bulgaricus*, *Bifidobacterium*, and *Streptococcus thermophilus*) could improve the degree of alcoholic liver injury in mice by inhibiting inflammation and adjusting gut microbiota. Gut microbiota and gut-liver axis play an important role in this, but specific substances or mechanisms remain to be investigated. In this study, 16SrDNA was used to analyze the gut microbiota of mice. Sun, et al. [41] results showed that alcohol induced decreased Alpha diversity in the gut microbiota of mice, but Alpha diversity was restored by pre-intake of all doses of *L. plantarum* J26. The results of Beta diversity showed that the alcohol-induced gut microbiota structure of mice was significantly different from the other groups, and *L. plantarum* J26 could adjust the structure of gut microbiota to gradually approach the NC group.

At the phylum level, the abundance of *Firmicutes* and *Verrucomicrobia* decreased significantly and the abundance of *Actinobacteria*, *Proteobacteria* and *Bacteroidetes* increased in the mice induced by alcohol. Many members of *Firmicutes* are beneficial bacteria, such as the common *Lactobacillus*. The decline in the abundance of *Verrucomicrobia* leads to a decrease in the abundance of its dominant bacterium *Akkermansia*, which was confirmed by the results of this study. However, the abundance of *Actinobacteria* and *Proteobacteria* increased in the presence of alcohol. The same result was obtained in Engen’s [42] study. *Actinobacteria* and *Proteobacteria* include many pathogens and can induce inflammation in the organism. Llopis et al. [43] studied the changes in the gut microbiota of liver-injured mice by alcohol feeding for 5 weeks and found that the relative abundance of *Bacteroidete* increased significantly, which was consistent with the results of the present study. *Bacteroidetes* were mainly composed of Gram-negative bacteria and closely related to LPS secretion. However, it has two sides: on the one hand, it can disturb the homeostasis of gut microbiota and cause endogenous infection [44]; on the other hand, the *Bacteroides fragilis* under its door can produce polysaccharide A to relieve colitis [45]. Compared to the Alcohol group, pre-intake of *L. plantarum* J26 reversed these changes to varying degrees, suggesting that the protective effect of *L. plantarum* J26 on gut microbiota could be related to anti-inflammatory activity.

At the genus level, alcohol induction significantly increased the abundance of *Aerococcus* and *Sutterella* in the gut microbiota of mice, which pre-intake of *L. Plantarum* J26 reversed this trend. *Aerococcus* is an important pathogenic bacterium in clinical practice, which can cause intestinal and extraintestinal infections in humans and is often present in different inflammasomes. It has been reported that *Aerococcus* is also the dominant genus of gut microbiota in chemical-induced liver injury [46]. *Sutterella* secretes excessive immunoglobulin A (IgA) protease, which reduces the concentration of IgA in intestinal mucosa and impairs the function of the intestinal antimicrobial immune response. In a set of in vitro experiments, *Sutterella* was found to adhere to intestinal epithelial cells and promote the secretion of IL-8 [47]. Compared to the Alcohol group, *L. plantarum* J26 was able to increase the abundance of *Allobaculum* in a dose-dependent manner. Studies have shown that *L. Plantarum* J26 could protect the intestinal barrier by increasing tryptophan metabolites [48]. In the study of Zeng et al. [49], it was also found that the addition of *Allobaculum* contributes to ameliorating gut microbiota imbalance and enhancing the intestinal barrier to alleviate alcohol-induced liver injury in mice. The abundance of *Akkermansia*, *Lactobacillus*, *Bifidobacterium*, *Oscillospira*, and *Ruminococcus* in the *L. plantarum* J26 group increased at different doses. The decrease of *Akkermansia* abundance is a characteristic of alcohol-induced gut microbiota structural imbalance. Studies in mice models and clinical trials have demonstrated that oral supplementation of *Akkermansia* can improve alcohol-induced liver injury, indicating the important role of *Akkermansia* in alcohol-induced liver injury. In addition, *Akkermansia* can increase the content of short-chain fatty acids (SCFAs), such as acetic acid and butyric acid in the intestine. *Lactobacillus* and *Bifidobacterium*, which are common probiotic genera, can inhibit the growth of harmful bacteria, produce acetic acid and propionic acid [50], and maintain intestinal health [51]. *Oscillospira* has never been cultured but has been continuously detected through the development of 16s rDNA technology. It is widely presented in animal and human intestines and has been listed as a candidate for next-generation probiotics due to its ability to produce butyric acid [52]. *Ruminococcus* is one of the most effective bacterial genera for carbohydrate decomposition, which has functions, such as stabilizing the intestinal barrier, reversing diarrhea, and reducing the risk of colorectal cancer [53]. These results suggested that pre-intake of *L. plantarum* J26 could restore alcohol-induced gut microbiota disturbance and maintain intestinal barrier function.

## 5. Conclusions

This study demonstrated that pre-intake of *L. plantarum* J26 alleviated alcohol-induced liver inflammation in a dose-dependent manner. Pre-intake of *L. plantarum* J26 restored the disturbed gut microbiota, repaired the intestinal barrier, prevented LPS from crossing the intestinal barrier through the intestinal-liver axis to the liver, and reduced the originally alcohol-induced elevated hepatic LPS levels. It attenuates the activation of the TLR4-mediated MAPK signaling pathway and thus reduces the generation and development of liver inflammation. Therefore, *L. plantarum* J26 could be used as a potential functional food ingredient or a microbiological agent to alleviate alcohol-induced liver inflammation.

## Figures and Tables

**Figure 1 nutrients-15-00190-f001:**
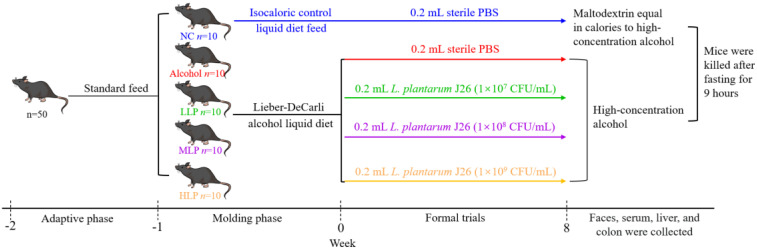
Scheme of animal experiment. NC, normal control; LLP, low-dose *L. plantarum* J26; MLP, medium-dose *L. plantarum* J26; HLP, high-dose *L. plantarum* J26.

**Figure 2 nutrients-15-00190-f002:**
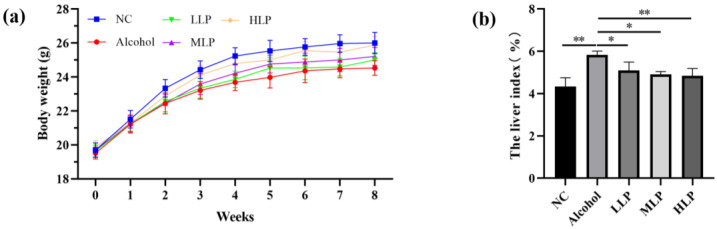
Effects of *L. plantarum* J26 on body weight and liver index in the mice. (**a**): body weight; (**b**): liver index. NC, normal control; LLP, low-dose *L. plantarum* J26; MLP, medium-dose *L. plantarum* J26; HLP, high-dose *L. plantarum* J26. The data were analyzed with a one-way ANOVA and expressed as means ± SD (*n* = 10). * *p* < 0.05, ** *p* < 0.01 compared with the Alcohol group.

**Figure 3 nutrients-15-00190-f003:**
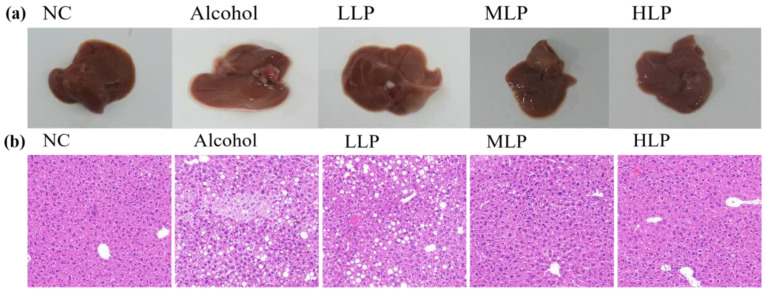
Effects of *L. plantarum* J26 on liver histopathology in liver of mice. (**a**): anatomical morphology of liver tissue; (**b**): hepatic histopathological alterations (H&E staining, magnification ×200). NC, normal control; LLP, low-dose *L. plantarum* J26; MLP, medium-dose *L. plantarum* J26; HLP, high-dose *L. plantarum* J26.

**Figure 4 nutrients-15-00190-f004:**
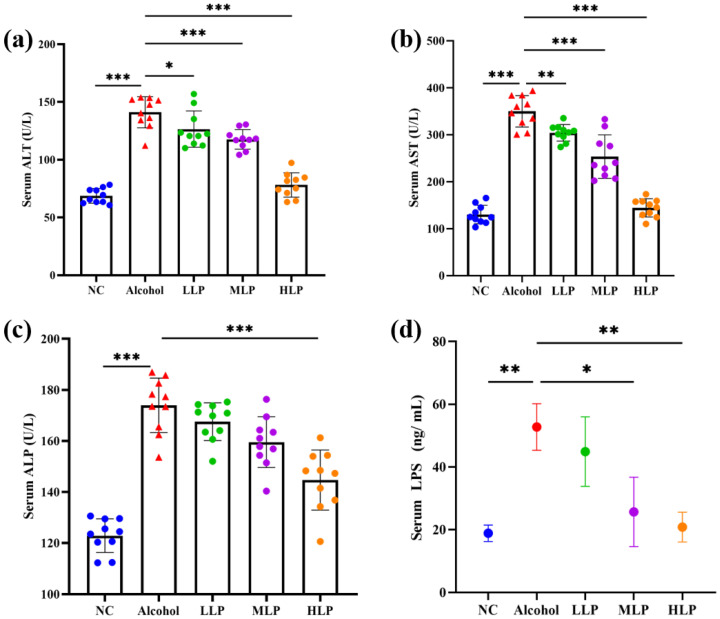
Effects of *L. plantarum* J26 on serum ALT, AST, ALP and LPS levels. (**a**): ALT activity; (**b**): AST activity; (**c**): ALP activity; (**d**): LPS content. ALT, alanine aminotransferase; AST, aspartate aminotransferase; ALP, alkaline phosphatase; LPS, lipopolysaccharide; NC, normal control; LLP, low-dose *L. plantarum* J26; MLP, medium-dose *L. plantarum* J26; HLP, high-dose *L. plantarum* J26. The data were analyzed with a one-way ANOVA and expressed as means ± SD (*n* = 10). * *p* < 0.05, ** *p* < 0.01 and *** *p* < 0.001 compared with the Alcohol group.

**Figure 5 nutrients-15-00190-f005:**
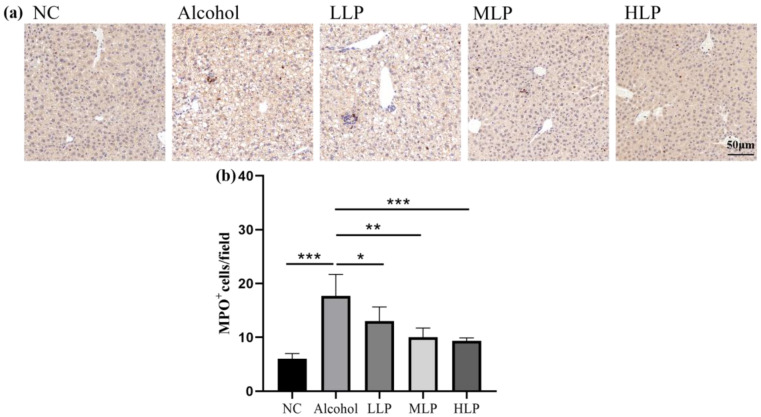
Effects of *L. plantarum J26* on the number of MPO-positive cells in livers. (**a**): immunohistochemical staining of MPO.; (**b**): MPO-positive cells. NC, normal control; LLP, low-dose *L. plantarum* J26; MLP, medium-dose *L. plantarum* J26; HLP, high-dose *L. plantarum* J26. The data were analyzed with a one-way ANOVA and expressed as means ± SD (*n* = 6). * *p* < 0.05, ** *p* < 0.01 and *** *p* < 0.001 compared with the Alcohol group.

**Figure 6 nutrients-15-00190-f006:**
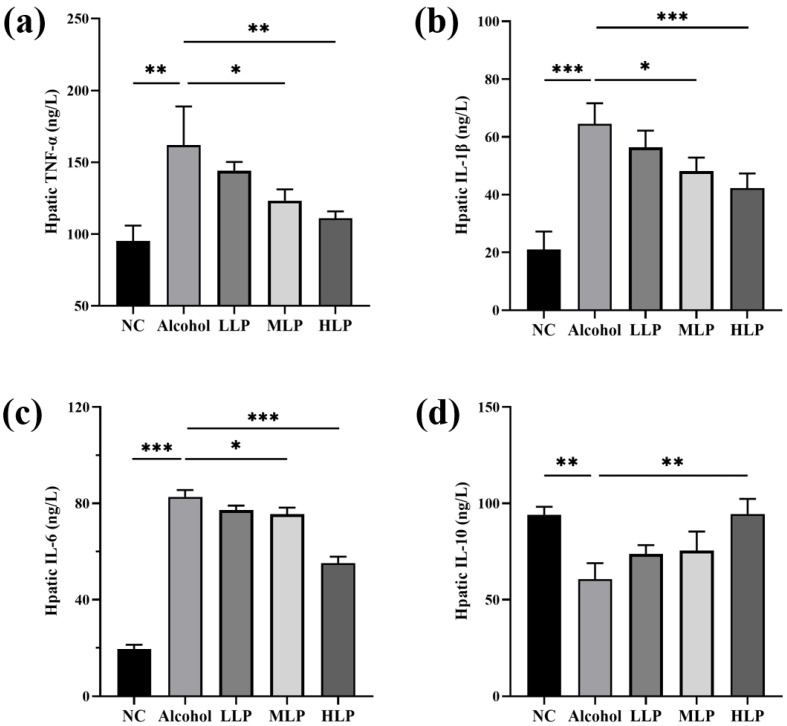
Effects of *L. plantarum* J26 on liver inflammatory factors. (**a**): TNF-α content; (**b**): IL-1β content; (**c**): IL-6 content; (**d**): IL-10 content. TNF-α, tumor necrosis factor-α; IL-1β, interleukin-1β; IL-6, interleukin-6; IL-10, interleukin-10; NC, normal control; LLP, low-dose *L. plantarum* J26; MLP, medium-dose *L. plantarum* J26; HLP, high-dose *L. plantarum* J26. The data were analyzed with a one-way ANOVA and expressed as means ± SD (*n* = 6). * *p* < 0.05, ** *p* < 0.01 and *** *p* < 0.001 compared with the Alcohol group.

**Figure 7 nutrients-15-00190-f007:**
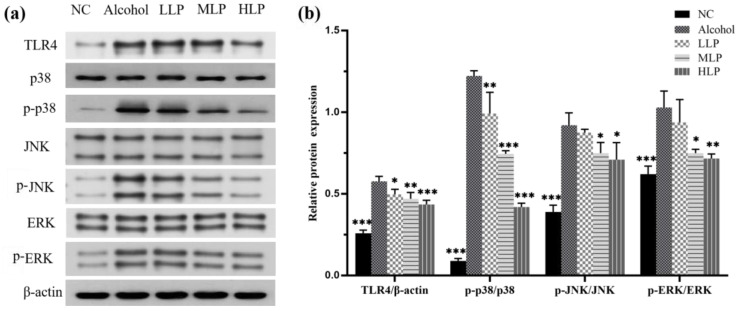
Effects of *L. plantarum* J26 on the expression of related gene proteins in TLR4 and MAPK signaling pathways. (**a**): the expression of TLR4, p38, *p*-p38, JNK, *p*-JNK, ERK, *p*-ERK protein in the liver; (**b**): Relative expression of each protein. NC, normal control; LLP, low-dose *L. plantarum* J26; MLP, medium-dose *L. plantarum* J26; HLP, high-dose *L. plantarum* J26. The data were analyzed with a one-way ANOVA and expressed as means ± SD (*n* = 3). * *p* < 0.05, ** *p* < 0.01 and *** *p* < 0.001 compared with the Alcohol group.

**Figure 8 nutrients-15-00190-f008:**
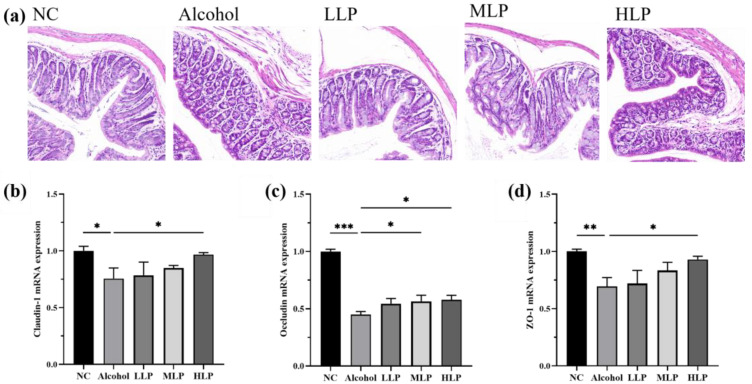
Effects of *L. plantarum* J26 on alcohol-induced intestinal injury in mice. (**a**): (H&E staining, magnification ×20; (**b**): gene expression of Claudin-1; (**c**): gene expression of Occludin; (**d**): gene expression of ZO-1. ZO-1, zonula occludens-1; NC, normal control; LLP, low-dose *L. plantarum* J26; MLP, medium-dose *L. plantarum* J26; HLP, high-dose *L. plantarum* J26. The data were analyzed with a one-way ANOVA and expressed as means ± SD (*n* = 6). * *p* < 0.05, ** *p* < 0.01 and *** *p* < 0.001 compared with the Alcohol group.

**Figure 9 nutrients-15-00190-f009:**
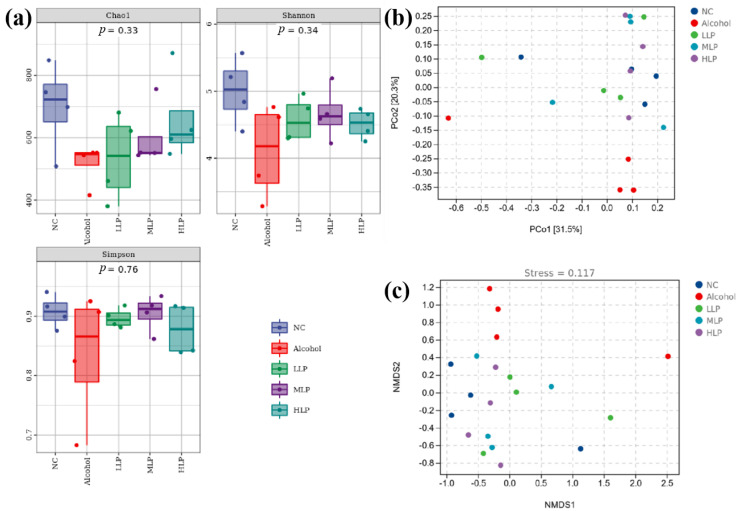
Effects of *L. plantarum* J26 on Microbial composition. (**a**): Alpha diversity index of intestinal microbiota; (**b**): PCoA analysis of psychrotrophic microbiome among samples; (**c**): NMDS analysis of groups. PCoA, principal coordination analysis; NMDS, non-metric multidimensional scaling; NC, normal control; LLP, low-dose *L. plantarum* J26; MLP, medium-dose *L. plantarum* J26; HLP, high-dose *L. plantarum* J26. The data were expressed as means ± SD (*n* = 4).

**Figure 10 nutrients-15-00190-f010:**
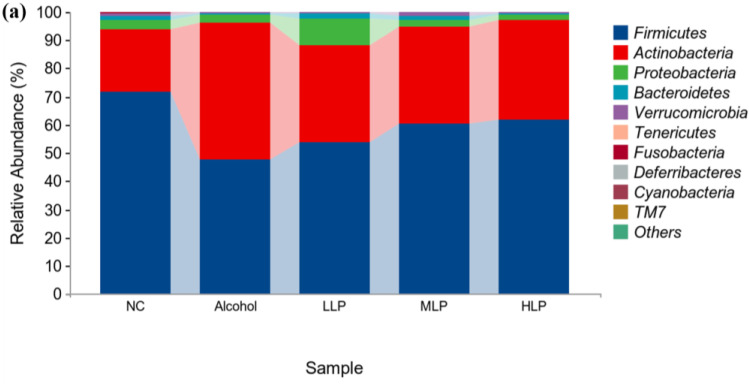
Effects of *L. plantarum* J26 on microbial composition of the cecal contents. (**a**): microbial composition at the phylum level of the cecal contents a; (**b**): microbial composition at the genus level of the cecal contents. NC, normal control; LLP, low-dose *L. plantarum* J26; MLP, medium-dose *L. plantarum* J26; HLP, high-dose *L. plantarum* J26.

**Table 1 nutrients-15-00190-t001:** The information on the primers.

Type	Gene Name	Base Sequence (5′ → 3′)
Tight junction protein	Claudin-1	F: AGATACAGTGCAAAGTCTTCGA R: CAGGATGCCAATTACCATCAAG
	Occludin	F: TGCTTCATCGCTTCCTTAGTAA R: GGGTTCACTCCCATTATGTACA
	ZO-1	F: CTGGTGAAGTCTCGGAAAAATG R: CATCTCTTGCTGCCAAACTATC

Note: F stands for forwarding sequence and R stands for reverse sequence. ZO-1, zonula occludens-1.

**Table 2 nutrients-15-00190-t002:** Lipid content in serum of mice.

Group	TG (mmol/L)	TC (mmol/L)
NC	0.93 ± 0.07 *	1.52 ± 0.20 **
Alcohol	1.25 ± 0.08	2.23 ± 0.08
LLP	1.10 ± 0.10	1.95 ± 0.19
MLP	1.08 ± 0.10	1.70 ± 0.13 *
HLP	0.89 ± 0.05 *	1.65 ± 0.05 *

Note: TG, triglyceride; TC, total cholesterol; NC, normal control; LLP, low-dose *L. plantarum* J26; MLP, medium-dose *L. plantarum* J26; HLP, high-dose *L. plantarum* J26. The data were analyzed with a one-way ANOVA and expressed as means ± SD (*n* = 10). * *p* < 0.05, ** *p* < 0.01 compared with the Alcohol group.

## Data Availability

Data in the project is still being collected, but all data used in the study is available by contacting the authors.

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
