# Peer review of "Lactobacillus plantarum J26 Alleviating Alcohol-Induced Liver Inflammation by Maintaining the Intestinal Barrier and Regulating MAPK Signaling Pathways"

_nutrients, 2022, doi:10.3390/nu15010190_

Round 1
Reviewer 1 Report
Lactobacillus plantarum J26 alleviating alcohol-induced liver inflammation by maintaining the intestinal barrier and regulating MAPK signaling pathways
Remarks to the Author:
The authors of this article have studied the important roles of pre-intake of Lactobacillus plantarum J26 (L. plantarum J26) in alcohol-induced liver inflammation. They found that L. plantarum J26 reduced the abundance of Gram-negative bacteria and maintained the intestinal barrier, thereby affecting reducing LPS circulation and gut-liver axis. Also, they found decreased TLR4-mediated MAPK signaling pathway, which is consistent with the phenotype. The manuscript is generally well-written. Also, the results are convincing, and the conclusions are appropriate. However, I have some concerns regarding specific points:
Specific comments:
1. The authors should conduct Oil-red O and TUNEL staining to clearly show the histopathologic effects (lipid accumulation and cell apoptosis).
2. Body weight, liver weight, and the liver weight to body weight ratio should be provided in Figure 2 or 3.
3. Serum or hepatic triglyceride and cholesterol should be determined.
4. In figure 6, NF-κB protein levels should be evaluated.
5. In figure 7, all markers (Claudin-1, Occludin, and ZO-1) should be determined by IHC or Western blotting.
6. Cell type-specific (at least hepatocyte and Kupffer cell) role of LPS-TLR4 should be more discussed in pathologic conditions.
7. Clinical studies (patients with liver diseases) concerning probiotics should be more discussed in the introduction or discussion part to improve the clinical relevance of this study.
Reviewer 2 Report
General Comments:
- Please use scientific words- using words like “extremely significant” is not appropriate. Usually significant differences with P-value is enough to confer the statistical differences. To highlight larger differences in mean, fold change can be used.
- Some of animal experiment design is not very clear. Schematic diagram helps, but text should be very crisp and meaningful in contextual manner. For example, what is Lieber-DeCarli alcohol liquid diet and what is its relevance in transition period should be explained with references.
- Abbreviations should be explained whenever first time is mentioned in the text.
- Methods should be re-written scientifically with many inconsequential things should be replaced with critical informations. For example, in western blot a detailed antibody citation is required with full product name, application, catalogue no, and species. Such things must follow standard scientific writing and presentation.
Major comments:
- The author is clearly over-signifying the results obtained in this study. It is very clear from Figure 5, that pro-inflammatory proteins are mostly significantly regulated by HLP, with only moderately effected by MLP and LLP treatment especially in IL-6 and IL-10. The author has to re-write these observations with only highlighting significant data.
- Similarly in Figurer 7 expression levels of related tight junction proteins Claudin-1, Occludin, and ZO-1 expression was only seen to be significantly modulated in HLP group, but author observed that even MLP group increased Occludin expression by 25.9% (looks like 10%), and there is no visible difference in Occludin expression between LLP and MLP treatment group . The author also has to re-write these observations with more accuracy and only highlighting the most significant results.
- A generally recommendation is that Asterisk symbol can be replaced by individual P-values (all treatment vs alcoholic group) even on graphs for better accuracy and consistency.
Round 2
Reviewer 1 Report
All issues have been addressed for publication in this journal.
Author Response
Thank the reviewer for the invaluable suggestions.